# Associations between serum pigment epithelium-derived factor and physical performance in older women: The Otassha study

Hiromichi Tsushima[1,2*], Takashi Shida[3], Sho Hatanaka[3], Takahisa Ohta[3], Narumi Kojima[3], Hiroyuki Sasai[3], Masataka Sugimoto[1,2*]

1 Molecular and Cellular Aging, Tokyo Metropolitan Institute for Geriatrics and Gerontology, Tokyo, Japan, 2 Department of Cellular Pathology, National Center for Geriatrics and Gerontology, Aichi, Japan, 3 Frailty and Musculoskeletal Health Research, Tokyo Metropolitan Institute for Geriatrics and Gerontology, Tokyo, Japan

* spt4527@gmail.com (HT); msugimot@ncgg.go.jp (MS)

## Abstract

Pigment epithelium-derived factor (PEDF) contributes to the beneficial effects of exercise by suppressing cellular senescence in multiple organs. Its expression declines in the skeletal muscle of aged animals, suggesting a role in age-related frailty. To investigate the association between circulating PEDF and physical performance, we analyzed its associations with skeletal muscle mass and mobility in a cohort of older women. A cross-sectional analysis was conducted in 143 community-dwelling females (mean age 77.01 ± 4.16 years). Serum PEDF and body composition were measured, and physical performance was assessed via gait speed, handgrip strength, leg extension strength, and the Five Times Sit-to-Stand test. Serum PEDF levels positively correlated with skeletal muscle mass index (SMI) and gait speed, while no significant correlation was observed between serum PEDF levels and other physical function parameters. These findings suggest that circulating PEDF is associated with skeletal muscle mass and mobility in older adults, warranting further investigation into its potential role as a myokine in humans.

## Introduction

Aging leads to declines in physical performance, including muscle strength and mobility, and is a major risk factor for chronic diseases, frailty, and dementia [1,2]. Physical inactivity, which becomes more common with age due to conditions such as sarcopenia (loss of muscle mass and strength) and chronic diseases, accelerates decline in mobility and increases susceptibility to health complications [3,4]. Therefore, promoting physical activity is essential for maintaining independence and quality of life in older adults [5,6].

**Data availability statement:** Full data is available in S1 Table of the manuscript.

**Funding:** JSPS KAKENHI 22K19743(MS) JSPS KAKENHI 22K16386 (HT) JSPS KAKENHI 24K02871 (MS) JSPS A3 JSPS = Japan Society for the Promotion of Science. Foresight Program JPJSA3F20230001 (MS) The funders had no role in study design, data collection and analysis, decision to publish, or preparation of the manuscript.

**Competing interests:** The authors have declared that no competing interests exist.

Senescent cells accumulate in various tissues with age in both humans and mice [7,8], as well as under pathological conditions [9]. These cells contribute to chronic diseases through their inflammatory secretions, collectively known as the senescence-associated secretory phenotype (SASP) [10]. Aerobic exercise has been shown to reduce cellular senescence in both human and rodent models [11,12], suggesting that targeting senescent cells may underlie the benefits of physical training.

We recently reported that Pigment Epithelium-Derived Factor (PEDF) mediates the beneficial effects of exercise in rodent models [13]. PEDF suppresses cellular senescence in cultured cells [14], and its expression in muscle tissue and circulating blood increases significantly following prolonged voluntary exercise training in mice [13,15]. Furthermore, PEDF expression in muscle is markedly reduced in aged (>20-month-old) animals compared to young (2-month-old) animals. These findings suggest that PEDF may act as a myokine/exerkine, mediating the anti-senescence effects of exercise.

Recent studies have also identified PEDF as an adipokine secreted by adipose tissue, where it modulates lipid metabolism and inflammation [16]. Collectively, these findings indicate that PEDF may function as both a myokine and an adipokine, mediating crosstalk between muscle and adipose tissues in the regulation of metabolic homeostasis. However, the relative contributions of these tissue sources to circulating PEDF levels remain to be elucidated.

Although PEDF's role in cellular senescence and muscle function has been demonstrated in experimental models, its relationship with physical performance and muscle mass in humans remains largely unexplored. Therefore, this study aimed to investigate whether circulating PEDF levels are associated with physical performance and muscle mass in older adults. We specifically hypothesized that circulating PEDF may reflect the combined influence of muscle- and adipose-derived activity, linking metabolic and functional status in aging. To test this hypothesis, we examined the relationship between circulating PEDF levels and physical performance, and muscle mass in a cohort of older Japanese women recruited at the Tokyo Metropolitan Institute for Geriatrics and Gerontology. Our findings suggest that circulating PEDF levels are associated with muscle mass and mobility in older adults, supporting its potential role as a myokine in human physiology.

## Materials and methods

### Study subjects

This study was approved by the Research Ethics Committee of the Tokyo Metropolitan Institute for Geriatrics and Gerontology (approval number: R21-16 and R22-048). A cross-sectional study was conducted in 2023 using samples from community-dwelling older adults residing in the Itabashi Ward of metropolitan Tokyo. Participants were recruited between 01/08/2023 and 31/08/2023. Of the 777 participants, 87.6% (681 individuals) were women. The study was explained to the participants using materials, and their written informed consent was obtained. Data on complete medical history were obtained using a questionnaire (yes/no). In addition, smoking habits (Current/Quit (for over a year)/Never), Alcohol drinking habits (Current/Quit (for

over a year)/Never), Exercise habits (per week) (Daily/5–6 days/2–4 days/1 day/None). Participants were further screened based on the following criteria: only women participants were included, as the number of men recruited in this community-dwelling cohort was very small, which precluded adequately powered analyses and could introduce sex-related confounding, blood samples were collected at least 6 hours after the last meal, completion of all physical function tests, and a maximum score on the Basic Activities of Daily Living scale. After applying these criteria, 143 participants remained eligible, and their blood samples were used for biochemical measurements and statistical analyses. An anonymized participant-level dataset is provided as Supporting Information (S1 Table).

## Anthropometric measurements

Height and weight were measured, and body mass index (BMI) was calculated as weight divided by height squared (kg/m²). Skeletal muscle mass index (SMI) and body fat percentage were assessed using the bioelectrical impedance analysis device (InBody 770, InBody Co. Ltd., South Korea) [17].

## Physical performance measurements

Physical performance measurements were conducted as previously described, assessing handgrip strength, leg extension strength, Gait speed, and the Sit-to-Stand test [17–20]. Grip strength was measured using a Jamar hydraulic grip-strength meter (SAKAI Medical Co., Ltd., Japan), with the higher value from two dominant hand trials recorded. Gait speed was measured over a 5-meter walking path, with an additional 6-meter auxiliary path to allow for acceleration and deceleration. Leg extension strength was assessed using a Load cell (LUX-B-1KN-ID, Kyowa Electric Instruments Co., Ltd., Japan), with participants seated on a horizontal chair, knees bent at 90°. The load cell was positioned 5 cm above the lateral malleolus, and maximal voluntary contractions were performed after training. In the Sit-to-Stand test, the time to complete five consecutive sit-to-stand cycles was measured.

## Enzyme-linked immunosorbent assay (ELISA)

Serum was obtained by centrifugation at 3,000 rpm for 15 min. The serum samples were stored at −80 °C until analysis. Serum PEDF levels were analyzed using a Human PEDF ELISA kit (RD191114200R, BioVendor R&D®). According to the manufacturers' data sheet, the intra-assay and inter-assay coefficients of variation were <5% and <7%, respectively. Optical density measurements were carried out using the VICTOR Nivo Multimode Microplate Reader (PerkinElmer Inc.).

## Statistical analysis

All statistical analyses were performed using IBM SPSS Statistics version 29 software (RRID: SCR_019096, IBM). Age-adjusted Pearson correlation coefficients were calculated to assess the relationships between serum PEDF concentrations and other variables. Age-adjusted analysis of covariance (ANCOVA) was conducted to evaluate the associations between serum PEDF concentrations and disease history.

Multiple regression analysis, adjusted for potential confounding factors (age, body fat percentage, and diabetes), was conducted to explore the relationships between serum PEDF concentrations and physical performance measures. Lifestyle assessments (exercise frequency, alcohol, and smoking habit) were not used as confounding factors in the regression analysis due to the limited number of samples. Although the sample size (n = 143) was relatively small, the participant-to-variable ratio exceeded 40:1, satisfying recommended criteria for regression stability. A post hoc power analysis, based on the observed effect sizes for the associations between lnPEDF and body composition measures, indicated approximately 80% power to detect moderate effects (α = 0.05). The assumptions for multiple regression analysis were verified by examining the normality of residuals (histogram and normal probability plots), linearity (scatterplots of standardized residuals versus predicted values), and multicollinearity (variance inflation factors, all < 2).

Quartile-based ANCOVA and Bonferroni post-hoc analysis were conducted to compare serum PEDF levels across groups stratified by gait speed and SMI. In all analyses, serum PEDF concentrations were log-transformed (lnPEDF) due to its non-normal distribution.

All analyzed variables are provided in the anonymized dataset available as Supporting Information (S1 Table).

## Results

### Study subject characteristics

Table 1 summarizes the age, medical history, lifestyle and physical performance characteristics of the study participants. The underlying anonymized participant-level data are available as Supporting Information (S1 Table). The mean age of the cohort was $77.01 \pm 4.16$ years. The participants had an average body fat percentage of $31.71 \pm 6.77\%$ and a mean SMI of $5.78 \pm 0.59\,kg/m^2$, reflecting typical age-related body composition changes.

Regarding medical history, 35.7% of participants reported hypertension, 7.0% had diabetes mellitus, and 25.9% had osteoarthritis of the knee, indicating the prevalence of chronic conditions in this older population. Lifestyle assessments showed that 59% of participants engaged in regular exercise (at least once a week), 35% were current drinkers, and 2.8% were current smokers.

Serum PEDF levels were assessed to determine their associations with body composition and physical performance. The mean serum concentration of PEDF was $13.08 \pm 3.3$ µg/mL. Physical performance metrics revealed a mean time of $1.31 \pm 0.18$ m/s for the Gait speed, indicative of functional mobility. Handgrip strength averaged $19.58 \pm 4.37$ kg, and leg extension strength was $162.52 \pm 55.16$ N, both aligning with typical values observed in older women. The average time for the 5 Times Sit-to-Stand test was $8.11 \pm 1.93$ seconds, reflecting the cohort's lower limb strength and endurance.

### Associations between serum PEDF levels and physical and medical parameters

The relationships between serum PEDF levels and various physical and body composition parameters are presented in Table 2. Serum PEDF levels (lnPEDF) positively correlated with body weight ($r=0.317$, $p<0.001$), BMI ($r=0.329$, $p<0.001$), SMI ($r=0.179$, $p<0.033$), and body fat percentage ($r=0.366$, $p<0.001$). These results suggest that PEDF levels are closely associated with body composition markers in this population, as shown in Fig 1. The correlation between skeletal muscle mass index and body fat percentage was moderate ($r=0.378$, $p<0.001$), indicating low collinearity between these two parameters.

Among participants, those with a history of hypertension or osteoarthritis had significantly higher age-adjusted mean lnPEDF concentrations than those without these conditions (2.60 vs. 2.51, $p=0.027$; 2.63 vs. 2.51, $p=0.007$, respectively; Table 3), which may reflect a link between PEDF and metabolic or inflammatory processes associated with these conditions.

### Association of serum PEDF with physical performance

The association between serum PEDF levels and physical performance measures was assessed using multiple regression analysis (Table 4). Serum PEDF levels were significantly associated with gait speed ($\beta=0.174$, $p=0.029$), suggesting that higher PEDF levels correlate with better walking performance. In contrast, no significant associations were found between serum PEDF levels and other performance metrics, including grip strength, leg extension strength, or the Five Times Sit-to-Stand test. This association between lnPEDF and gait speed was also observed in the quartile analysis (S2 Table). These findings suggest that serum PEDF may be more closely linked to functional mobility, as reflected by gait speed, rather than overall muscle strength or endurance, supporting its potential role in maintaining mobility and physical function in older adults.

**Table 1. Study subjects (*n* = 143).**

| Characteristic | Mean (*SD*) |
|---|---|
| Age (y) | 77.01 (4.16) |
| Height (cm) | 151.60 (5.21) |
| Weight (kg) | 51.37 (7.93) |
| Body mass index (kg/m$^2$) | 22.34 (3.23) |
| Skeletal muscle mass index (kg/m$^2$) | 5.78 (0.59) |
| PEDF (µg/mL) | 13.08 (3.30) |
| Percent Body Fat (%) | 31.71 (6.77) |
| Gait speed (m/s) | 1.31 (0.18) |
| Handgrip strength (kg) | 19.58 (4.37) |
| Leg extension strength (N) | 162.52 (55.16) |
| Five Times Sit-to-Stand test (s) | 8.11 (1.93) |
| Medical history | (%) |
| Hypertension | 35.7 |
| Diabetes mellitus | 7.0 |
| Stroke | 1.4 |
| Heart disease | 11.9 |
| Dyslipidemia | 52.4 |
| Osteoporosis | 22.4 |
| Respiratory disease | 6.3 |
| Collagen disease | 4.2 |
| Osteoarthritis of the knee | 25.9 |
| Fracture | 2.8 |
| Ear disease | 10.5 |
| Cancer | 7.7 |
| Parkinson's disease | 0.0 |
| Depression | 2.8 |
| Alzheimer's disease | 0.0 |
| Vascular dementia | 0.0 |
| Other types of dementia | 0.0 |
| Mild cognitive impairment | 0.0 |
| Other brain diseases | 3.5 |
| Alcohol drinking habits | |
| Current | 35.0 |
| Quit (for over a year) | 6.3 |
| Never | 58.7 |
| Smoking habits | |
| Current | 2.8 |
| Quit (for over a year) | 11.2 |
| Never | 86.0 |
| Exercise habits (per week) | |
| Daily | 13.3 |
| 5–6 days | 15.4 |
| 2–4 days | 24.5 |
| 1 day | 5.6 |
| None | 41.3 |

**Table 2. Correlation between PEDF levels and physical parameters (*n* = 143).**

| Variable | Correlation | |
|---|---|---|
| | *r* | *p* |
| Height (cm) | 0.024 | 0.781 |
| Weight (kg) | 0.317 | <0.001 |
| Body mass index (kg/m²) | 0.329 | <0.001 |
| Skeletal muscle mass index (kg/m²) | 0.179 | 0.033 |
| Percent Body Fat (%) | 0.366 | <0.001 |
| Gait speed (m/s) | −0.108 | 0.168 |
| Handgrip strength (kg) | 0.084 | 0.319 |
| Leg extension strength (N) | 0.143 | 0.089 |
| Five Times Sit-to-Stand test (s) | 0.068 | 0.419 |

Age-adjusted Pearson's correlation coefficients between lnPEDF and each parameter.

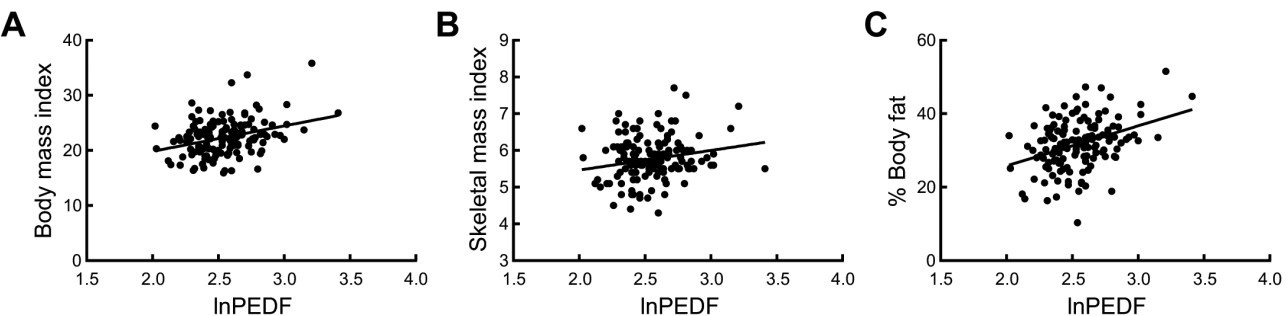

**Fig 1. Correlations between serum lnPEDF concentrations and body composition parameters.** Scatter plots showing the correlations between serum lnPEDF concentrations and body composition parameters. Serum lnPEDF was positively associated with (A) body mass index, (B) skeletal muscle mass index, and (C) percent (%) body fat. Regression lines are shown for reference. Corresponding correlation coefficients and *p*-values are presented in Table 2.

**Table 3. Disease history of the study subjects (*n* = 143).**

| Disease | Hypertension | | Diabetes mellitus | | Osteoarthritis | |
|---|---|---|---|---|---|---|
| Disease History | Yes | No | Yes | No | Yes | No |
| n | 51 | 92 | 10 | 133 | 37 | 106 |
| lnPEDF Mean *SD* | 2.60 *0.26* | 2.51 *0.20* | 2.69 *0.32* | 2.53 *0.22* | 2.63 *0.25* | 2.51 *0.21* |
| p value | 0.027 | | 0.071 | | 0.007 | |

Age-adjusted mean lnPEDF concentrations were analyzed by disease history (Yes/No) for hypertension, diabetes mellitus, and osteoarthritis. Data were analyzed using age-adjusted ANCOVA.

## Discussion

In mice, muscle PEDF expression declines significantly with age, potentially linking it to age-related changes in physical activity and muscle function. In this study, we found that circulating PEDF levels positively correlated with skeletal muscle mass and gait speed in older women, suggesting an association between PEDF and physical function. While PEDF acts as a myokine/exerkine that suppresses cellular senescence in mice [13], its role as a myokine in humans remains to be

**Table 4. Association of PEDF levels with physical performance and skeletal muscle mass index is multiple regression analysis (n = 143).**

| Variable | β | 95% confidence interval | | p |
|---|---|---|---|---|
| | | lower limit | upper limit | |
| Gait speed | 0.174 | 0.024 | 0.429 | 0.029 |
| Handgrip strength | 0.090 | −0.003 | 0.013 | 0.258 |
| Leg extension strength | 0.129 | 0.000 | 0.001 | 0.106 |
| Five times Sit-to-Stand test | 0.003 | −0.019 | 0.019 | 0.971 |

Adjusted for age, Percent Body Fat, and Diabetes mellitus.

determined. PEDF has been shown to exert anti-inflammatory, antioxidant, and anti-senescent effects in various tissues. In skeletal muscle, PEDF may help preserve mitochondrial function and reduce oxidative stress, thereby maintaining muscle quality and contractile capacity. In vascular and metabolic tissues, its anti-inflammatory actions could improve perfusion and metabolic efficiency, which in turn may indirectly contribute to better mobility in older adults. However, the observed correlations may reflect that PEDF serves as a biomarker associated with muscle homeostasis and mobility status rather than a causal factor. Longitudinal or interventional studies will be required to clarify the directionality of these associations.

Notably, these associations were more pronounced in individuals with lower gait speed or SMI, suggesting that PEDF may be particularly relevant in conditions of impaired physical function. Nevertheless, despite its association with gait speed, no significant correlations were observed between PEDF levels and other physical performance measures, such as grip strength, leg extension strength, or the Five Times Sit-to-Stand test. This distinction suggests that PEDF may be more closely associated with mobility than with overall muscular strength.

Our findings also demonstrate a significant positive correlation between serum PEDF levels and body fat. This relationship suggests that PEDF may reflect metabolic or physiological states associated with body composition, aligning with prior research indicating that PEDF may also function as an adipokine [16] and contribute to lipid metabolism [21,22]. Indeed, PEDF is one of the most abundant proteins secreted by adipocytes and has been shown to modulate insulin sensitivity and inflammatory signaling in fat and muscle cells [16]. Therefore, circulating PEDF levels may primarily reflect adipose tissue metabolism rather than solely muscle-derived activity, supporting the notion that PEDF functions as a systemic regulator of metabolic homeostasis. Additionally, these results suggest that circulating PEDF levels may be influenced not only by muscle but also by adipose tissue, highlighting its potential role in metabolic regulation and muscle function in aging populations.

Participants with a history of hypertension and osteoarthritis exhibited higher serum PEDF levels compared to those without these conditions. These findings may suggest associations between PEDF and vascular or bone diseases, consistent with previous reports of its roles in vascular remodeling and bone metabolism [23–26]. Elevated PEDF levels in hypertension and osteoarthritis could be indicative of underlying inflammation or metabolic dysregulation associated with these conditions. Given that both hypertension and osteoarthritis are characterized by chronic low-grade inflammation, the elevated PEDF levels observed in these participants may reflect a compensatory response to systemic inflammatory stress. These findings indicate that PEDF elevation in such conditions may primarily reflect systemic inflammatory activity rather than muscle-specific secretion, supporting its role as a circulating mediator that links inflammation and metabolic regulation in older adults. Similarly, prior studies have reported an association between elevated PEDF levels and diabetes [24,27]. We also observed a trend toward higher PEDF levels in participants with diabetes, suggesting a potential role of PEDF in the metabolic dysregulation associated with diabetes. Furthermore, PEDF has been implicated in neurodegenerative conditions such as Alzheimer's disease [28,29]. Although none of the participants had dementia, long-term follow-up studies are needed to determine whether PEDF levels may be associated with the onset of these chronic diseases.

In summary, our study suggests that PEDF is associated with body composition, chronic disease states, and functional mobility in older women. Circulating PEDF levels were positively correlated with walking performance and skeletal muscle mass, with stronger associations observed among individuals with lower gait speed or reduced muscle mass. Because of the cross-sectional design, these findings should be interpreted as associations rather than causal relationships. Future longitudinal or interventional studies are needed to clarify whether changes in PEDF levels influence, or are influenced by, physical performance. Additionally, the potential role of PEDF as an exerkine and its possible involvement in senescence suppression in aging populations warrant further investigation. This study has several limitations. First, inflammatory markers such as C-reactive protein were not measured, which limits our ability to directly relate PEDF levels to systemic inflammation. Second, the participants were all women recruited from a single center, and thus the findings may not be generalizable to men or other populations. Third, because the participants were community-dwelling volunteers, selection bias toward relatively healthy older adults cannot be excluded. In addition, the relatively high proportion of participants with hypertension and osteoarthritis may have influenced circulating PEDF levels, which should be considered and controlled for in future studies.

## Supporting information

**S1 Table. Anonymized participant-level data used in this study.**
(XLSX)

**S2 Table. Relationship between Quartiles of Gait Speed and Skeletal Muscle Mass Index with Circulating lnPEDF Levels Adjusted for Potential Confounder.**
(DOCX)

## Acknowledgments

We thank Dr. Yasunori Fujita and Dr. Mikako Hirose for their assistance.

## Author contributions

**Conceptualization:** Masataka Sugimoto.

**Data curation:** Hiromichi Tsushima.

**Funding acquisition:** Hiromichi Tsushima, Masataka Sugimoto.

**Investigation:** Hiromichi Tsushima, Takashi Shida, Sho Hatanaka, Takahisa Ohta, Narumi Kojima, Hiroyuki Sasai.

**Methodology:** Takashi Shida, Hiroyuki Sasai.

**Project administration:** Masataka Sugimoto.

**Resources:** Takashi Shida, Sho Hatanaka, Takahisa Ohta, Narumi Kojima, Hiroyuki Sasai.

**Writing – original draft:** Hiromichi Tsushima.

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
