## [Decision Letter · Decision Letter 0]

26 Oct 2025

Dear Dr. Tsushima,

Thank you for submitting your manuscript to PLOS ONE. After careful consideration, we feel that it has merit but does not fully meet PLOS ONE’s publication criteria as it currently stands. Therefore, we invite you to submit a revised version of the manuscript that addresses the points raised during the review process.

We look forward to receiving your revised manuscript.

Kind regards,

Hidetaka Hamasaki

Academic Editor

PLOS ONE

Journal Requirements:

JSPS KAKENHI 22K19743(MS)

JSPS KAKENHI 22K16386 (HT)

JSPS KAKENHI 24K02871 (MS)

JSPS A3 Foresight Program JPJSA3F20230001 (MS)

3. Please expand the acronym “JSPS” (as indicated in your financial disclosure) so that it states the name of your funders in full.

We thank Dr. Yasunori Fujita and Dr. Mikako Hirose for their assistance. This work was supported by JSPS KAKENHI (22K19743, 22K16386 and 24K02871), JSPS A3 Foresight Program (JPJSA3F20230001), and internal operational grants by Tokyo Metropolitan Institute for Geriatrics and Gerontology.

JSPS KAKENHI 22K19743(MS)

JSPS KAKENHI 22K16386 (HT)

JSPS KAKENHI 24K02871 (MS)

JSPS A3 Foresight Program JPJSA3F20230001 (MS)

5. Please provide a complete Data Availability Statement in the submission form, ensuring you include all necessary access information or a reason for why you are unable to make your data freely accessible. If your research concerns only data provided within your submission, please write "All data are in the manuscript and/or supporting information files" as your Data Availability Statement.

6. Please remove all personal information, ensure that the data shared are in accordance with participant consent, and re-upload a fully anonymized data set.

Reviewers' comments:

Reviewer's Responses to Questions

**Comments to the Author**

1. Is the manuscript technically sound, and do the data support the conclusions?

Reviewer #1: Partly

Reviewer #2: Yes

2. Has the statistical analysis been performed appropriately and rigorously?

Reviewer #1: Yes

Reviewer #2: Yes

3. Have the authors made all data underlying the findings in their manuscript fully available?

Reviewer #1: Yes

Reviewer #2: Yes

4. Is the manuscript presented in an intelligible fashion and written in standard English?

Reviewer #1: Yes

Reviewer #2: Yes

Reviewer #1: This manuscript investigates the relationship between circulating PEDF levels and physical performance in community-dwelling older women. The topic is timely and of potential significance for understanding the role of PEDF as a myokine related to aging and muscle function. The manuscript is clearly written, methodologically sound, and the findings are generally well supported by the data.

However, several issues should be addressed to improve the clarity, methodological transparency, and interpretability of the study.

Introduction

The introduction is concise and logically structured. However, the research hypothesis and biological rationale could be more explicitly stated. For example, clarify whether the study aims to determine if circulating PEDF acts primarily as a muscle-derived or adipose-derived factor influencing mobility.

Some recent literature discussing PEDF’s dual role as a myokine and adipokine could be better integrated to contextualize the study’s novelty.

The final paragraph should clearly articulate the specific objectives and the main hypothesis to guide readers.

Materials and Methods

The study design and inclusion criteria are appropriate; however, the rationale for including only female participants should be explained.

The description of physical activity and lifestyle assessments (exercise frequency, alcohol, smoking) is detailed, but it would be useful to mention whether these variables were considered as potential covariates in the regression analysis.

The ELISA method for PEDF quantification is standard, but please provide details on assay reproducibility (intra-/inter-assay coefficients of variation).

In the statistical analysis section, the selection of covariates (age, body fat, diabetes) appears somewhat limited. Given PEDF’s known associations with inflammation, vascular function, and metabolic diseases, additional potential confounders—such as hypertension, osteoarthritis, or activity level—should be considered or at least discussed as limitations.

State explicitly whether regression assumptions (normality, linearity, multicollinearity) were verified.

Consider adding a brief statement on sample size justification or post hoc power analysis, as n=143 may be modest for multivariate modeling.

Results

The results are well organized and tables are clear. However, consider presenting a scatter plot to visually support the main correlations.

Table 2 shows positive correlations between PEDF and both SMI and body fat percentage. Since PEDF is produced by both skeletal muscle and adipose tissue, it would be valuable to comment on potential collinearity between these variables.

The authors note that participants with hypertension and osteoarthritis had higher PEDF levels. Given that these conditions involve chronic inflammation, this observation could be relevant to the interpretation of PEDF’s systemic roles; including this point in the Discussion would strengthen the argument.

Discussion

The Discussion appropriately interprets the findings, but the cross-sectional nature of the study must be more explicitly emphasized as a limitation. The current wording could give the impression of causality (“PEDF contributes to…”). Please revise to indicate that associations do not establish directionality.

The positive correlation between PEDF and body fat percentage should be discussed in more depth. It remains unclear whether PEDF primarily reflects adipose tissue metabolism rather than muscle-derived activity.

Consider expanding the discussion on biological mechanisms: how PEDF may influence mobility (e.g., through anti-inflammatory or anti-senescent pathways).

The observation of higher PEDF levels in hypertension and osteoarthritis should be more fully integrated into the physiological interpretation, as these may reflect systemic inflammation rather than muscle-specific effects.

Limitations should include: (1) lack of inflammatory markers (e.g., CRP), (2) single-sex and single-center sample, and (3) possible selection bias toward relatively healthy older adults.

General and Language Comments

The manuscript is overall well written in clear English. Only minor grammatical refinements are needed:

Abstract: “To explore the relationship…” → “To investigate the association…”

Methods: “all of medical history” → “complete medical history”

Discussion: “stronger associations observed in individuals with impaired conditions” → “stronger associations observed among individuals with lower gait speed or reduced muscle mass.”

Reviewer #2: This study brings to the Fore a critical marker that helps us to understand physical performance with aging and the study was presented in a very simple, yet enligthened way.

The onlt suggestion i have is for the authors to possibly outline the potential for participants having hypertension and Osteoarthritis, having a higher PEDF, affecting the generlizability of the results. this should be stated as a limitation of the study with the reccomendation that this will be controlled for in future studies.

**Do you want your identity to be public for this peer review?** For information about this choice, including consent withdrawal, please see our Privacy Policy

Reviewer #1: No

Reviewer #2: No

---

## [Author Response · Author response to Decision Letter 1]

10 Nov 2025

Response to reviewers

PONE-D-25-13518

Associations between Serum Pigment Epithelium-Derived Factor and Physical Performance in Older Women: The Otassha Study

>Introduction

>The introduction is concise and logically structured. However, the research hypothesis and biological rationale could be more explicitly stated. For example, clarify whether the study aims to determine if circulating PEDF acts primarily as a muscle-derived or adipose-derived factor influencing mobility.

Thank you for this insightful comment.

We agree that the research hypothesis and biological rationale should be more clearly stated. We have revised the final paragraph of the Introduction to explicitly describe the hypothesis that circulating PEDF may reflect both muscle- and adipose-derived activity and could influence physical performance through these tissue interactions.

>Some recent literature discussing PEDF’s dual role as a myokine and adipokine could be better integrated to contextualize the study’s novelty.

Thank you for this insightful comment.

We agree that recent findings describing the dual role of PEDF as both a myokine and adipokine are relevant to contextualize our study. We have now revised the Introduction to integrate this concept and to highlight the novelty of examining circulating PEDF in relation to both muscle and adipose contributions to physical function in older adults.

>The final paragraph should clearly articulate the specific objectives and the main hypothesis to guide readers.

Thank you for this valuable suggestion.

We have revised the final paragraph of the Introduction to explicitly state the study objectives and the main hypothesis, clarifying that the study aimed to examine whether circulating PEDF levels are associated with physical performance and muscle mass in older adults.

>Materials and Methods

>The study design and inclusion criteria are appropriate; however, the rationale for including only female participants should be explained.

Thank you for this helpful comment. Both sexes were eligible for inclusion; however, in this community-dwelling cohort of older adults the number of men recruited was very small, which precluded adequately powered multivariable analyses. To avoid unstable estimates and sex-related confounding, we therefore analyzed data from women only and have clarified this in the Materials and Methods. We acknowledge this as a limitation and note that validation in male cohorts is warranted.

>The description of physical activity and lifestyle assessments (exercise frequency, alcohol, smoking) is detailed, but it would be useful to mention whether these variables were considered as potential covariates in the regression analysis.

Thank you for this helpful comment.

We agree that lifestyle factors such as exercise frequency, alcohol consumption, and smoking status are important variables that may influence PEDF levels. However, given the relatively small sample size (n = 143) and the primary focus on metabolic conditions previously linked to PEDF, we limited the number of covariates in the regression model to avoid model overfitting. Among these variables, diabetes was considered a key covariate based on prior evidence of its association with PEDF. We acknowledge that lifestyle factors may also contribute to PEDF variability, and their inclusion should be addressed in larger, future studies.

>The ELISA method for PEDF quantification is standard, but please provide details on assay reproducibility (intra-/inter-assay coefficients of variation).

Thank you for this helpful comment.

We have added information on assay reproducibility in the Materials and Methods section. According to the manufacturer’s data sheet for the ELISA kit (BioVendor, RD191114200R), the intra-assay and inter-assay coefficients of variation were <5% and <7%, respectively, indicating high assay reproducibility.

>In the statistical analysis section, the selection of covariates (age, body fat, diabetes) appears somewhat limited. Given PEDF’s known associations with inflammation, vascular function, and metabolic diseases, additional potential confounders—such as hypertension, osteoarthritis, or activity level—should be considered or at least discussed as limitations.

Thank you for this valuable comment.

We agree that additional confounding factors such as hypertension, osteoarthritis, and activity level may influence PEDF levels. We have expanded the Discussion to include these points and noted in the Limitations section that these conditions could have affected circulating PEDF levels and should be controlled for in future studies.

>State explicitly whether regression assumptions (normality, linearity, multicollinearity) were verified.

Thank you for this valuable comment.

We have verified that the assumptions for multiple regression analysis were met. Specifically, normality of residuals was confirmed by histogram and normal probability plots, linearity was checked using scatterplots of standardized residuals versus predicted values, and multicollinearity was assessed using variance inflation factors (all VIF < 2). We have now stated this explicitly in the Statistical Analysis section.

>Consider adding a brief statement on sample size justification or post hoc power analysis, as n=143 may be modest for multivariate modeling.

Thank you for this valuable comment.

Although the sample size (n=143) may appear modest, the main variables associated with serum lnPEDF showed moderate correlations, which correspond to medium effect sizes. A post hoc power analysis based on these observed effects indicated approximately 80% power (α = 0.05) for detecting associations of this magnitude. We have now added a brief statement describing this in the Statistical analysis section.

>Results

The results are well organized and tables are clear. However, consider presenting a scatter plot to visually support the main correlations.

We thank the reviewer for this helpful suggestion.

In the revised manuscript, we have added a new figure (Figure 1) showing scatter plots illustrating the correlations between serum lnPEDF concentrations and key body composition parameters, including BMI, SMI, and body fat percentage. This figure provides a visual representation of the main correlations reported in Table 2.

>Table 2 shows positive correlations between PEDF and both SMI and body fat percentage. Since PEDF is produced by both skeletal muscle and adipose tissue, it would be valuable to comment on potential collinearity between these variables.

Thank you for this valuable comment.

We assessed potential collinearity between these parameters, and found that their correlation was moderate (r = 0.378, p < 0.001), indicating low collinearity between them. We clarified this point in the Results.

>The authors note that participants with hypertension and osteoarthritis had higher PEDF levels. Given that these conditions involve chronic inflammation, this observation could be relevant to the interpretation of PEDF’s systemic roles; including this point in the Discussion would strengthen the argument.

Thank you for this valuable comment.

We agree that the higher PEDF levels observed in participants with hypertension and osteoarthritis may reflect the chronic inflammatory status associated with these conditions. PEDF is known to exert anti-inflammatory and vasculoprotective effects, and its elevation could represent a compensatory response to low-grade systemic inflammation. We have now incorporated this point into the Discussion to provide a broader context for PEDF’s systemic roles.

>Discussion

>The Discussion appropriately interprets the findings, but the cross-sectional nature of the study must be more explicitly emphasized as a limitation. The current wording could give the impression of causality (“PEDF contributes to…”). Please revise to indicate that associations do not establish directionality.

Thank you for this valuable comment.

We agree that the cross-sectional design precludes any inference of causality. We have now revised the Discussion to clarify that the observed associations do not establish directionality, and that longitudinal or interventional studies are required to determine causal relationships between PEDF and physical performance.

>The positive correlation between PEDF and body fat percentage should be discussed in more depth. It remains unclear whether PEDF primarily reflects adipose tissue metabolism rather than muscle-derived activity.

Thank you for this important comment.

We agree that the positive correlation between PEDF and body fat percentage warrants further discussion. Previous studies have shown that PEDF is abundantly secreted by adipocytes and regulates lipid metabolism and inflammatory signaling in adipose tissue. Therefore, circulating PEDF may partly reflect adipose tissue–derived activity rather than muscle-derived secretion. We have now expanded the Discussion to emphasize that serum PEDF likely represents the integrated output of multiple tissues, including adipose tissue, and may act as a systemic metabolic regulator rather than a muscle-specific myokine.

>Consider expanding the discussion on biological mechanisms: how PEDF may influence mobility (e.g., through anti-inflammatory or anti-senescent pathways).

Thank you for this helpful comment.

We agree that further discussion of the potential biological mechanisms linking PEDF and mobility would improve the manuscript. We have now expanded the Discussion to include possible pathways through which PEDF may influence physical function, including its anti-inflammatory and anti-senescent effects on muscle and vascular tissues.

>The observation of higher PEDF levels in hypertension and osteoarthritis should be more fully integrated into the physiological interpretation, as these may reflect systemic inflammation rather than muscle-specific effects.

Thank you for this valuable comment.

We agree that the higher PEDF levels observed in participants with hypertension and osteoarthritis may reflect systemic inflammation rather than muscle-specific effects. We have revised the Discussion to more fully integrate this point into the physiological interpretation, emphasizing that elevated PEDF levels could represent a compensatory response to chronic inflammatory stress and metabolic dysregulation rather than a direct marker of muscle activity.

>Limitations should include: (1) lack of inflammatory markers (e.g., CRP), (2) single-sex and single-center sample, and (3) possible selection bias toward relatively healthy older adults.

Thank you for this helpful comment.

We agree with the reviewer’s points regarding the study limitations. We have now expanded the final paragraph of the Discussion to acknowledge the absence of inflammatory markers such as CRP, the single-sex and single-center nature of the sample, and the potential selection bias toward relatively healthy older adults.

>General and Language Comments

>The manuscript is overall well written in clear English. Only minor grammatical refinements are needed:

>Abstract: “To explore the relationship…” → “To investigate the association…”

>Methods: “all of medical history” → “complete medical history”

>Discussion: “stronger associations observed in individuals with impaired conditions” → “stronger associations observed among individuals with lower gait speed or reduced muscle mass.”

Thank you for these helpful language suggestions. The indicated phrases in the Abstract, Methods, and Discussion have been revised accordingly.

>Reviewer #2: This study brings to the Fore a critical marker that helps us to understand physical performance with aging and the study was presented in a very simple, yet enligthened way.

The onlt suggestion i have is for the authors to possibly outline the potential for participants having hypertension and Osteoarthritis, having a higher PEDF, affecting the generlizability of the results. this should be stated as a limitation of the study with the reccomendation that this will be controlled for in future studies.

Thank you for this positive evaluation and constructive suggestion. We agree that the higher prevalence of hypertension and osteoarthritis among participants may affect the generalizability of the findings. We have now added a statement in the Limitations section noting that this factor should be considered and controlled for in future studies.

---

## [Decision Letter · Decision Letter 1]

30 Nov 2025

Associations between Serum Pigment Epithelium-Derived Factor and Physical Performance in Older Women: The Otassha Study

PONE-D-25-13518R1

Dear Dr. Tsushima,

We’re pleased to inform you that your manuscript has been judged scientifically suitable for publication and will be formally accepted for publication once it meets all outstanding technical requirements.

Kind regards,

Hidetaka Hamasaki

Academic Editor

PLOS ONE

Additional Editor Comments (optional):

Reviewers' comments:

Reviewer's Responses to Questions

**Comments to the Author**

Reviewer #2: All comments have been addressed

2. Is the manuscript technically sound, and do the data support the conclusions?

Reviewer #2: Yes

3. Has the statistical analysis been performed appropriately and rigorously?

Reviewer #2: Yes

4. Have the authors made all data underlying the findings in their manuscript fully available?

Reviewer #2: Yes

5. Is the manuscript presented in an intelligible fashion and written in standard English?

Reviewer #2: Yes

Reviewer #2: ll corrections have been implemented, The recommendations were fully adhered to and the body of work us ready for publication.

**Do you want your identity to be public for this peer review?** For information about this choice, including consent withdrawal, please see our Privacy Policy

Reviewer #2: No

---

## [Editor Report · Acceptance letter]

PONE-D-25-13518R1

PLOS One

Dear Dr. Tsushima,

I'm pleased to inform you that your manuscript has been deemed suitable for publication in PLOS One. Congratulations! Your manuscript is now being handed over to our production team.

Kind regards,

on behalf of

Dr. Hidetaka Hamasaki

Academic Editor

PLOS One